# Verification of the Influence of Loading and Mortar Coating Thickness on Resistance to High Temperatures Due to Fire on Load-Bearing Masonries with Clay Tiles

**DOI:** 10.3390/ma12223669

**Published:** 2019-11-07

**Authors:** Rodrigo P. de Souza, Fernanda Pacheco, Gustavo L. Prager, Augusto M. Gil, Roberto Christ, Vinícius Muller de Mello, Bernardo F. Tutikian

**Affiliations:** 1PPGEC, UNISINOS, São Leopoldo, RS 93022-750, Brazil; pericorodrigo@gmail.com; 2Civil Engineer Program, PPGEC/Itt Performance, UNISINOS, São Leopoldo, RS 93022-750, Brazil; fernandapache@unisinos.br (F.P.); gprager@unisinos.br (G.L.P.); rchrist@unisinos.br (R.C.);; 3Department of Civil and Environmental Engineering, Michigan State University, East Lansing, MI 48824, USA; masierog@msu.edu; 4Departamento de Civil y Ambiental, Universidad de la Costa, Barranquilla 080020, Colombia

**Keywords:** structural masonry, mortar coating, fire resistance

## Abstract

Masonry has been widely used as a construction method. However, there is a lack of information on its fire behavior due to the multitude of variables that could influence this method. This paper aimed to identify the influence of loading and mortar coating thickness on the fire behavior of masonry. Hence, six masonries made of clay tiles laid with mortar were evaluated. The mortar coating had a thickness of 25 mm on the face not exposed to high temperatures, while the fire-exposed face had thicknesses of 0, 15, and 25 mm. For each mortar coating thickness, two specimens were tested, with and without loading of 10 tf/m. The real-scale specimens were subjected to the standard ISO 834 fire curve for four hours, during which the properties of stability, airtightness, and thermal insulation were assessed. Results showed that loaded specimens yielded smaller deformations than unloaded ones. Samples that lacked mortar coating on the fire-exposed face underwent fire resistance decrease of 27.5%, while the ones with 15 mm decreased by 58.1%, and the ones with 25 mm decreased by 41.0%. As mortar coating thickness increased, the plane deformations decreased from 40 mm to 29 mm and the thermal insulation properties of the walls improved significantly. For specimens with mortar coating thickness of 25 mm, the load application resulted in a reduction of 23.8% of the thermal insulation, while the unloaded specimen showed a decrease of 43.3%, as well as a modification of its fire-resistance rating.

## 1. Introduction

When fire strikes a structural masonry building, it endangers the building’s physical structure and the life of its occupants [1]. A fire usually develops in four stages: incipient, growth, fully developed, and decay, abiding by a typical temperature-time curve during this process [2]. The thermal action from fire conditions is described by radiation and convection heat transfer mechanisms, whereas the heat from the fire can spread by conduction, convection, and radiation [3]. Several technical standards specify conditions for safe building operations and even include exceptional events such as fires [4]. There are also functional safety requirements concerning fire prevention and extinguishing, ways to limit fire spread and prevent damages and even collapse, along with safe user evacuation methods and safe access routes for rescue teams and firefighters [5,6].

Compartment walls and floors are part of these considerations, as they pose as fire-resistant elements that strive to stop fire from spreading to adjacent rooms [7]. According to Al-Hadhrami and Ahmad [8], masonry walls made with clay bricks can be used for horizontal compartmentalization and as firewalls. As per Nguyem and Meftah [9], the fire resistance of masonry walls can be determined either by laboratory tests or by semi-empirical methods that are more conservative. The fire resistance of masonry walls covers three safety levels represented by structural stability, air and smoke tightness, and thermal insulation, according to [10], as well as the main standards related to the fire protection test [11,12,13,14]. Gomes-Heraz et al. [15] pointed out that heating with real flames is the most realistic approach to analyze the fire resistance of various materials. Regarding structural stability, it evaluates the preservation of mechanical strength, maintaining the safe distribution of loads to the foundation [16]. Airtightness refers to the capacity of the wall to prevent the passage of flames and hot gases through cracks that may appear [17]. For Ono [18], thermal insulation is the criterion that evaluates if the heat on the fire-exposed surface of the building element poses a risk to the people and the objects occupying rooms that are adjacent to the unexposed face.

As for thermal resistance, clay bricks rely on the specific gravity of their constituting materials. Moreover, masonry with hollow cores hinder heat transfer [4]. According to the Brick Industry Association (BIA) [19], the cavities of clay bricks make heat transfer happen through convection and radiation. The internal temperature distribution profile in fire resistance tests suggests that temperature distribution is not linear because it is influenced by the thickness of the wall and the heating rate [20,21]. The concavity of the curve generated by thermal action is higher for thinner walls and it increases along with the heating rate [20]. Considering heat storage and dissipation, the main factors that can influence the thermal inertia of a wall are: the thickness of the wall, specific gravity, specific heat capacity, and the thermal conductivity of the materials. Beall [22] states that thermal inertia contributes to the fire resistance of masonry, as it hinders heat transfer.

When subjected to high temperatures, masonry tends to bend towards the fire. According to Ono [18], stability is more relevant for load-bearing masonry. The degradation of materials exposed to high temperatures, added to differential thermal expansion, may bring the element to collapse depending on the deterioration degree [20]. As per Ayala [23], there are other critical factors that strive to maintain stability such as the heating ratio, exposure time, and moisture conditions. The degradation of materials after exposure to high temperatures, the bending resulting from thermal expansion, the restrictions, the geometry of the wall, and the presence of eccentricities from the strength lost by the fire-exposed face are interconnected [16]. Figure 1 depicts the arching towards the side with higher temperature, which is typical of clay masonry.

A wall that has been subjected to compressive axial stress shall yield second order effects that can be reduced during the initial stages of load application. Vertical loading tends to cause problems after a long period of exposure as the element tends to become laterally unstable due to the steep stresses produced by the moments from lateral displacement [24]. Nevertheless, Nadjaii, O’Gara and Ali [25] managed to perform numerical simulations of a masonry wall to analyze the influence of the eccentricity of load application, the slenderness of the wall, and the type of binding. In terms of loading in fire conditions, non-load bearing masonry presents displacements on both axes around it, with greater deformation at the center of the sample, behaving as if it were fixed on the four corners [3]. As for load-bearing masonry, the displacements are smaller and there is a restraint at the top and the base of the masonry, as depicted in Figure 2. In conclusion, loading has direct effect on the global displacement of masonry.

Nguyen and Meftah [9] linked the simultaneous occurrence of explosive spalling and vertical loads to the increased bending moment, owing to the eccentricity of the load applied. Spalling creates a zone of brittleness that allows warping and affects the sample as a whole, and local failure is an important factor for the performance of masonries in fire conditions. Load-bearing masonry walls undergo progressive compressive strength loss due to the deterioration of the mortar coating at high temperatures [26,27,28]. Experimental studies revealed that, upon receiving loads, the walls mitigated the increase of deformations. Silva, Oliveira and Sobrinho [29] observed that applying 30 mm thick mortar coating to both faces of prisms could increase load-bearing capacity by over 300%. Mortar coating reduces bending in comparison to uncoated samples. Still, there are few studies that provide background on the performance of load-bearing masonries with respect to fire. Therefore, this study aimed to assess the effect of mortar coating and loading on load-bearing walls made of clay tiles subjected to high temperatures.

## 2. Materials and Methods

The experimental procedure comprised of six fire resistance tests, as shown in Table 1.

### 2.1. Sample Composition

All walls were made up of clay tiles laid with mortar, with 1 mm thick vertical and horizontal joints. All samples had their unexposed face coated with 25 mm thick mortar. On the fire-exposed face, the coating was null for the reference walls, and 15 mm or 25 mm for the others, in order to evaluate the influence of mortar on fire resistance. The walls were executed within metal frames with internal span of 3.15 m × 3.00 m and curing lasted 56 days. The materials used during the investigation were cement type III from ASTM C150 [30], two types of silica sand whose particle-size distributions are presented in Table 2 in accordance with EN 933-1 [31], structural clay tiles with dimensions of 14 cm × 19 cm × 29 cm (width, length, and height) and properties presented in Table 3, characterized as per ASTM C-652 [32], and traditional mortar for laying and coating. The fresh and hardened state properties of the coating mortar were assessed, with the results being shown in Table 4.

### 2.2. Sample Instrumentation

#### 2.2.1. Measurement of Displacements

The displacements were measured with a Leica TS15 Robotic Total Station with an angular accuracy of ±3″ and linear accuracy of ±(1 mm + 1.5 mm km^−1^), placed on a metal tripod 6 m away from the unexposed face. After these measurements, the Cloud Compare software was used to process the 3D point cloud. According to Georgantas, Bredif, and Pierrot-Desseilligny [33], Cloud Compare offers several alternatives to measure the distance between two dot clouds, two meshes, or even between a cloud and a mesh. This study measured two specific clouds before and during the test. The software program then generated a local model for a reference point cloud and granted total and local accuracy over the distance calculated between the two-point clouds to measure the displacements.

#### 2.2.2. Temperature Measurement

Temperature evolution was monitored continuously during the test, considering that two thermocouples were installed inside the furnace to measure internal temperature. During the test, the temperature of the exposed face is the arithmetic mean of the temperatures measured by five thermocouples symmetrically arranged within the oven, one at the center of the sample and the others at the center of each quadrant, minding the same positioning on the unexposed face. Besides the internal and external thermocouples, 10 thermocouples were added along the cutting of the clay tiles and the coating, as depicted in Figure 3. The thermocouples were inserted in holes that had been drilled on the wall. It was not necessary to repair the wall because the hole and the thermocouple had similar dimensions, and it was assumed that adding other materials to these spots would alter the measurements. These data made it possible to define the temperature profile, assessing the influence of coating thickness and thermal inertia of the masonry. The thermocouples were installed on the 55th day of curing along the thickness of the wall, in its inner core and the center of two tiles, according to the following distances:First position (d = 0 cm), face of the tile exposed to fire regardless of the coating;Second position (d = 3 cm), 3 cm away from the fire-exposed face plus coating thickness;Third position (d = 7 cm), axis of the tile;Fourth position (d = 11 cm), 4 cm away from the axis of the tile;Fifth position (d = 14 cm), 7 cm away from the axis of the tile, on the external face of the tile.

The instrumentation relied on the use of type K thermocouples with 6 mm of diameter to measure the temperature within the furnace and on the fire-exposed face. The thermocouples along the thickness of the wall and on the unexposed surface were type K with 1.5 mm of diameter. The thermocouples on the external face were type T and were attached to copper disks with 12 mm of diameter and 0.5 mm of thickness. The procedure was complemented with a FLIR thermographic camera model A320 with a resolution of 320 × 240 pixels, sensitive to wavelengths of 7.5 and 13 µm, with a temperature gradient of 0 °C to 350 °C and a thermal resolution of ±2 °C, in order to analyze the temperature distribution and make it easier to identify airtightness. Moreover, the results presented in item 3 were obtained by the interpolation of temperature measurement spots that were counted with thermocouples.

### 2.3. Fire Resistance Tests

The test apparatus comprised a vertical furnace to which the sample was attached as per Figure 4. It had four burners that used liquefied petroleum gas as fuel, two on each side, without direct contact from the flames on the sample, applied according to the standard ISO 834-1 curve [10]. This apparatus abides by standardized procedures and has been used in other research as well [1,34,35]. The horizontal displacement was measured by a mesh of 49 points, with spacing of 50 cm between each one, covering an area of 9.45 m^2^. The measurements were performed every 10 min with the use of the total station equipment. The acceptable limit for horizontal displacement, set by ISO 834 [10], is L/30, where L is the length of constraining extremities. In this study, L was equal to 3150 mm, so the horizontal deflection limit was 105 mm. Every fire resistance test lasted 240 min, except for masonry W2, with a loading of 10 tf/m and no coating on its internal face, as it failed after 102.5 min of testing.

## 3. Results

### 3.1. Sample W1—Unloaded, No Internal Coating

At 5 min, the sample presented a vertical crack on its center. At the eighth minute, the first signs of smoke appeared on the vertical crack. After the airtightness test, it was noted that the sample remained airtight. At 11 min, another crack appeared, and then the wall retained its integrity up unril 4 h of testing had occurred. The temperature evolution of wall W1 is depicted in Figure 5. It shows through an exponential equation that the curvature was reduced. Figure 6 then used these data to depict the temperature evolution in the coated clay tiles.

The maximum displacement during the test period was 41 mm, on the fire-exposed face, which occurred at 150 min. It took place in the center of the plane tested, depicted by the blue hue, 35 mm at 30 min (Figure 7a) and 41 mm at 150 min (Figure 7b). Nguyen and Meftah [9] had carried out tests on clay bricks with a compressive strength of 8 MPA and no loading, and found a displacement of 40 mm on the exposed face at 30 min. According to these authors, the double bending of the external surface occurs due to differential thermal expansion, which provokes a tensile stress on the exposed face and tends to make it bend towards the fire.

In summary, the sample did remain stable despite the noteworthy passage of hot gases and smoke through the cracks on the unexposed face, which still did not set the cotton wad ablaze, as per procedures determined by ISO 834 [10]. Regarding thermal insulation, the external temperature increased and reached an arithmetic mean of 247.5 °C at 240 min, with a maximum punctual temperature of 288.1 °C.

### 3.2. Sample W2—Reference Sample, Loaded, No Internal Coating

In the first five minutes of testing, the sample presented no change whatsoever. Then, it presented a vertical crack on its center, on the middle one-third of its height. At 52 min, a horizontal crack appeared (Figure 8a) and, at 102 min, the wall lost its stability (Figure 8b).

The temperature evolution followed the same trend of exponential curve W1, since both walls had the same clay tiles and coating on the unexposed face (Figure 9).

The wall underwent higher displacement towards the furnace of 32 mm at 100 min. The displacements over time were lower than those of W1, so it can be assumed that the loading of 10 tf/m made the wall bend less, due to the greater binding to the test frame. Nguyen and Meftah [9] found lateral displacements of the plane of 23 mm at 30 min of testing, 3 mm smaller than the ones of this study, and 28 mm at 60 min, the same value, as they analyzed uncoated samples loaded with 13.2 tf/m. In terms of temperature, the maximum average was 96.6 °C and the punctual maximum was 105 °C (at 102 min).

### 3.3. Sample W3—Unloaded, Internal Coating of 15 mm

Sample W3 presented a vertical crack on its central part, located on the middle one-third, at 5 min of testing. At the sixth and ninth minutes, respectively, a horizontal crack appeared to the left of the sample, and so did an oblique crack on the lower edges. The sample remained airtight up until 240 min had passed. The temperature evolution measured by the thermocouples is shown in Figure 10, measured at 240 min of testing.

It is noteworthy that the temperature gradient became more intense with the passing of time due to the internal coating of 15 mm, which is the only difference between W1 and W3. Figure 11 then used these data to depict the evolution of the temperature of the coated ceramic tile.

When comparing the temperature evolution of walls W1 and W3, the mortar coating of 15 mm on the exposed face hindered the heating of the tiles. Between 30 min and 60 min there was no significant difference for the unexposed face, as the temperature varied by 9.9 °C. The difference stood out at 120 and 240 min, when the temperature of the external face showed reductions of 72.7 and 42.4 °C, respectively. The decreases in percent were 49.35% at 120 min and 17.13% at 240 min. The maximum displacement during the test was 43 mm on the exposed face at 30 min (Figure 12a), while the displacement outside the plane was stabilized at 120 min, varying 2 mm until reaching 240 min. Initially, the wall bended towards the interior of the furnace, but it backed off during the test without returning to its original position. At 240 min, the final displacement of the plane was 30 mm on the exposed face (Figure 12b), while that the maximum displacement throughout the entire test took place in the center of the plane, which is represented by the blue hue.

Comparing the displacements of W3 and W1 (Figure 6 and Figure 11), the curvature was similar for both walls, despite sample W3 presenting lower values due to the presence of reinforcement of 15 mm on the exposed face, which reduced the final lateral displacement by 10 mm at 240 min. The airtightness of the system was acceptable. In the end, concerning the thermal insulation criterion, there was an increase of the external temperature, reaching the arithmetic mean of 205.1 °C, with a punctual maximum of 228.8 °C.

### 3.4. Sample W4—Loaded, Internal Coating of 15 mm

At 11 min of testing, sample W4 presented oblique cracks on the lower corners and a vertical crack on its center (Figure 13a), similar to the behavior of W2 and W3. The oblique cracks appeared on the corners due to the greater binding and the consequent lower displacement. At 17 min, the first signs of smoke arose from the vertical crack (Figure 13b). The temperature evolution is depicted in Figure 14, which details the 240 min of testing.

Nadjai et al. [20] stated that load-bearing samples tend to deform and present vertical cracks, on both central area and middle point of the edges, thereby agreeing with the behavior observed in this study (Figure 13a).

As the test went on, the temperature variation along the thickness decreased over time. Based on the results of thermal gradient and thermographic camera, it was noted that the temperature of the clay tile was higher in the first few minutes, compared to the other coated samples, owing to the spalling of the coating on the inner side, of 15 mm, which allowed heat to act directly over the substrate. The values obtained were close to those pertaining to sample W1, with no internal coating. At 30 min, the first signs of spalling came about, as shown by the orange coloring on the top right side of Figure 15a, which was proven when the sample was detached from the furnace (Figure 15b). First there was the spalling of the coating, and then the chipping of the clay tile.

According to Nguyen and Meftah [9], chipping is a consequence of second order effects that result from the change of eccentricity of the wall, considering that the materials of the exposed face, clay tile and mortar, lose their mechanical properties. The wall initially bends towards the fire, and, during the test, it returns to the unexposed side, although it does not return to the starting position. After 240 min of testing, the final displacement of the plane was 20 mm to the unexposed side on the upper half of the sample, behavior similar to that of W2 but with higher values, since W2 had no mortar coating on the inner side.

In the experiment of Nguyen and Meftah [9], the sample underwent plane displacement of 14 mm towards the exposed side at 125 min. According to these authors, the simple bending of the external face was a result of differential thermal expansion, as tensile stresses act on the exposed side and provoke the bending towards the fire, despite the restraint on the top and bottom supports. Nevertheless, the sample presented stable behavior and remained airtight throughout the entirety of the test, even with displacements of 21 mm at 90 min. As for thermal insulation, there was an increase to the external temperature, with arithmetic mean of 344.7 and punctual maximum of 363.6 °C.

### 3.5. Sample W5—Unloaded, Internal Coating of 25 mm

During the first 10 min, nothing happened to sample W5. Then it presented oblique cracks on the lower corners, where displacement was smaller due to the restraint. At 60 min, small stains appeared around the sample, which resulted from moisture. Their appearance was explained by the noteworthy coating thickness of 25 mm on the exposed side, which hindered the sample’s loss of water. Throughout the 240 min of the fire resistance test, there was no evidence of smoke or hot gases being expelled from the cracks. The temperature measured at 120 min by the average of five points was 78.8 °C. The temperature evolution measured by the thermocouples of W5 is shown in Figure 15, after 240 min of testing.

The variation of the temperature along the thickness was smaller due to the coating of 25 mm on the exposed face. Wall W3 behaves similarly, despite its coating being 10 mm thinner. Figure 16 uses these results to depict the temperature evolution of the coated clay tile. Figure 17 shows the Temperature distribution of the tile at different times of exposure.

Comparing the evolution of temperature along the thickness of samples W1, W3, and W5, it can be stated that the mortar coating of 25 mm on the exposed face hindered the heating of the clay tile. In the first 60 min, the unexposed face presented a temperature difference of 9.6 °C, although the difference became more significant at 120 and 240 min as decreased by 68.5 and 114.6 °C, respectively. In other words, the temperature decrease on the unexposed face was 46.51% at 120 min and at 46.30% at 240 min.

The maximum displacement of wall W5 was 39 mm on the exposed face at 30 min of testing, which remained stable from 120 min to the end of the test at 240 min, when the final displacement of the plane was 29 mm. The displacements bore by W5 were similar to those of W1 due to the presence of coating of 25 mm on the exposed face, with a reduction of 11 mm in the final displacement, going from 40 mm to 29 mm.

During the 240 min of testing, the sample presented stable behavior. Regarding the airtightness of the system, there was no evidence of the passage of hot gases or smoke to the exterior of the wall through the oblique cracks. As for thermal insulation, there was an increase of external temperature, as it reached the average of 132.9 °C with a punctual maximum of 156.2 °C.

### 3.6. Sample W6—Loaded, Internal Coating of 25 mm

At 16 min, oblique cracks appeared on the lower corners, similar to the behavior of the other samples. At 57 min, small stains appeared around the sample due to moisture. This phenomenon was also identified during the test of W5 at 60 min. This happened because the walls had mortar coating of 25 mm on the exposed side, which hindered the loss of exceeding water during the 56 days of curing. At 80 min, the spalling of the mortar on the inner side was noted, as Figure 18a depicts. There were noteworthy differences in the color hues captured by the thermographic camera, which were later proved by Figure 18b. The temperature evolution measured by the thermocouples attached to W6 is shown in Figure 19, recorded after 240 min of testing.

Comparing the evolution of temperature along the thickness of walls W2, W4, and W6, it is possible to conclude that the mortar coating of 25 mm on the exposed face hindered the heating of the clay tile. In the first 60 min, the axis of the wall presented a reduction of 75.58%, going from 426.8 °C to 104.2 °C, that is, a decrease of 322.6 °C. In spite of the difference of 10 mm in the internal coating of W4 and W6, the difference on the external face was 176 °C, going from 344.7 °C to 168.7 °C, representing a decrease of 51.06% on the unexposed face. Therefore, the coating affected the performance of the wall in a positive way.

W6 presented maximum displacement of 24 mm towards the exposed side at 180 min. The displacement stabilized at 120 min and remained so until the end of the test at 240 min, when the final displacement of the plane was 23 mm towards the furnace. Wall W6 presented the smallest displacements compared to the other walls due to the combination of its thickness of 25 mm and the effect of loading, which restrained the displacement of the wall.

During the 240 min of testing, W6 presented stable behavior as it retained its structural stability. Regarding the airtightness of the system, there were no signs of the passage of gases or smoke to the outer side of the wall through the oblique cracks. As for thermal insulation, it was noted that the external temperature increased and achieved a mean of 168.7 °C with a punctual maximum of 231.5 °C.

## 4. Discussion

The comparative analysis between samples is based on Table 5, which summarizes the manifestations during the tests.

The analysis with a focus on the deformations of the samples shows that the values were higher for unloaded samples, when comparing the same coating and the absence of load. This behavior has already been reported by the literature [25], indicating that non-load beaning walls tend to present greater central bending, in contrast with load-bearing samples that present vertical deformations on the central axis and on the central point of each of their sides. Concerning the loading, its influence was noted when comparing samples W1 with W2, W3 with W4, and W5 with W6, bearing in mind that higher fire-resistance ratings (FRR) were attained by unloaded samples, along with higher displacement. Figure 20 shows the distribution of temperature over time for uncoated samples with variation in terms of load application.

Samples W1 and W2 demonstrate the relevance of having the fire-exposed face of a load-bearing masonry coated, due to the structural failure of sample W2 at 102 min. Sample W1, which was also uncoated but not loaded, performed better as it retained its structural stability up to 240 min. This event is possibly connected to the direct exposure of clay to temperatures close to those of its production process, as its microstructure changes at temperatures above 950 °C [24]. Such changes may have compromised the load-bearing capacity of the sample, as it bore load beyond its own weight and failed. Figure 21 presents the influence of loading on samples with a coating of 15 mm.

It is notable, mainly for the thermocouples attached to the inner face of the tiles (d = 0) that temperature development is more intense on load-bearing samples. Such a fact arises from the higher degree of cracking or even spalling on the masonry owing to the aforementioned changes in microstructure and loading. Figure 22 demonstrates that the increased coating and the consequent increased thermal inertia allowed the tiles to reach lower temperatures, which in turn had less impact, smoothing the effect of loading as the temperatures that change load-bearing capacity were not reached.

Figure 23 compares two extreme samples regarding the use of coating, that is, W1 with no coating, and W5 with coating of 25 mm on the fire-exposed face, in order to analyze unloaded samples.

From this figure, it is possible to assert that all depths of analysis attained improvements with regards to the attenuation of the maximum temperature. The impact was more critical for depths of 0 and 3 cm, with respect to the inner face of the tile. The improvement of the unexposed face was 17% and 46%, with coatings of 15 mm and 25 mm, respectively. According to ACI TMS 216.1 [36], adding a coating of 19 mm of cement and sand mortar over the screen, a procedure similar to the one performed on W4, should provide a gain of 20 min to fire resistance in comparison with the uncoated sample (W2). Moreover, adding 25.4 mm under the same conditions should increase this benefit to 30 min. The values obtained in this study surpass that expectation, since the addition of 15 mm compared to the uncoated sample provided an increase from 81 to 142 min of fire resistance, and from 81 to 221 min with 25 mm.

## 5. Conclusions

The experimental procedure results led to the conclusion that the unloaded sample that achieved the best performance with respect to the fire-resistance rating (FRR) was W5 (25 mm of coating on both faces), with an FRR of 240 min, namely, meeting the requirements of airtightness, structural stability and thermal insulation within this timespan. The samples tended to undergo decreased lateral displacements of the plane as their total thickness increased.

The loaded sample that showed the best performance for FRR was W6 (25 mm of coating on both faces), meeting the requirements of structural stability and airtightness while losing its thermal insulation at 221 min. Thus, coating thickness improved the performance of the masonry during the fire resistance test, as the temperature on the external face decreased by 46.3%, going from 247.5 °C to 132.9 °C on W5. W6 presented a decrease of 51.05% after the addition of 10 mm of coating on its exposed face, going from 15 mm to 25 mm.

Comparing same sample types with and without loading, it can also be concluded that the unloaded samples were the ones to present maximum displacements between 43 and 39 mm during the fire resistance test. This became clear as there were no restraints on the top and bottom parts, hence allowing the plane to bend with double curvature. On the other hand, the loaded samples underwent deformations on the upper part with values between 26 and 23 mm, most likely due to the restraint of the top and bottom edges caused by the loading, hence forcing the simple curvature bending.

This study had exploratory and explanatory natures, and was developed because the market lacks standards that guide the design of this type of system in these conditions, whereas the already-existing guidelines do not address load-bearing masonries and the wide range of compositions involved. By performing tests on different types of samples, it was possible to note peculiar behaviors among the typologies evaluated. Both the presence of loading and mortar coating thickness turned out to have an influence on the fire resistance of load-bearing masonry systems. It was noted that only sample W2 underwent a loss of stability and airtightness (wall with loading and no internal coating) as well failing at 102 min, hence forcing the sudden end of the test. The event of spalling caused by the loss of the internal part of the clay tiles was detected during the test with the aid of the thermographic camera. This manifestation was predominant for load-bearing samples. Also, the presence of loading does influence the direction towards which the sample bends.

## Figures and Tables

**Figure 1 materials-12-03669-f001:**
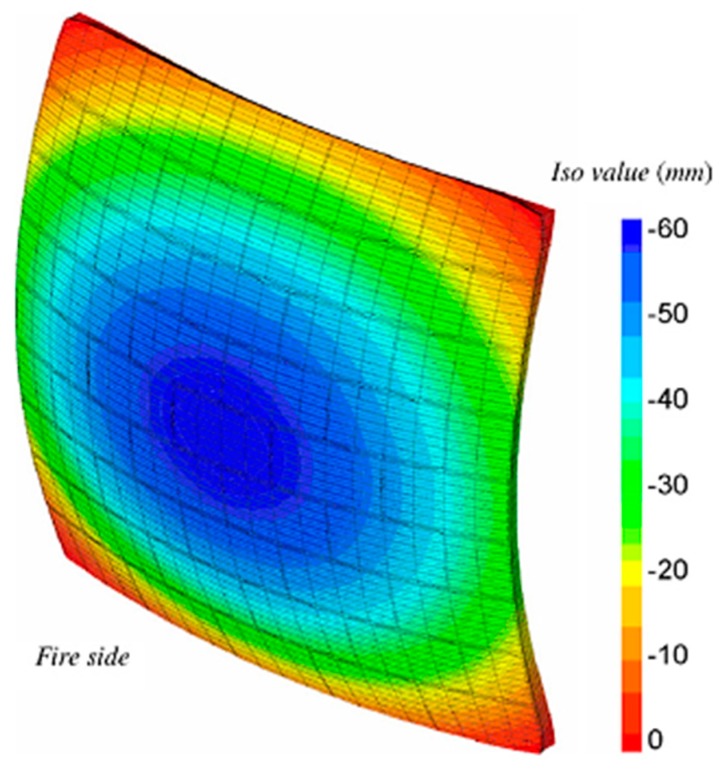
Displacement of the wall nearing the heat source [21].

**Figure 2 materials-12-03669-f002:**
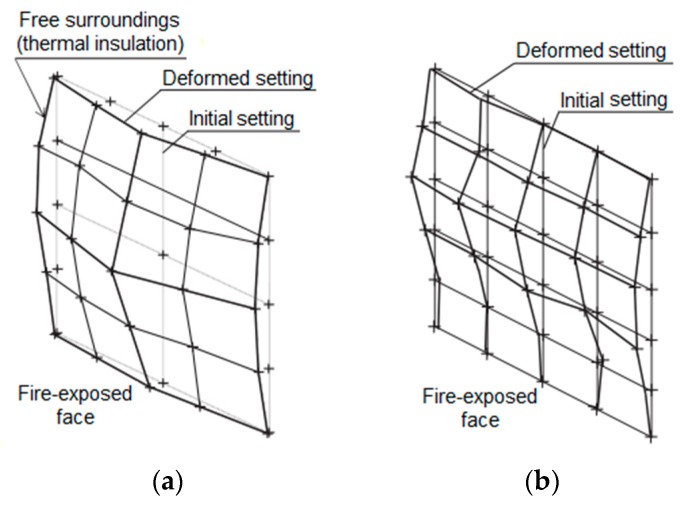
Deformations endured by fire-exposed masonry (**a**) non-load bearing wall and (**b**) load-bearing wall, as per Nguyen and Meftah [9].

**Figure 3 materials-12-03669-f003:**
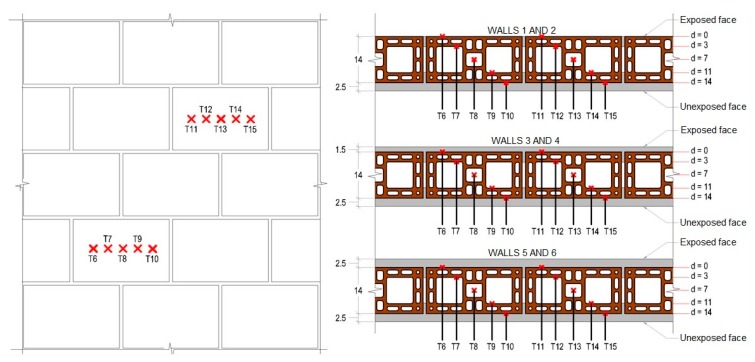
Thermocouple placement along the thickness of the wall.

**Figure 4 materials-12-03669-f004:**
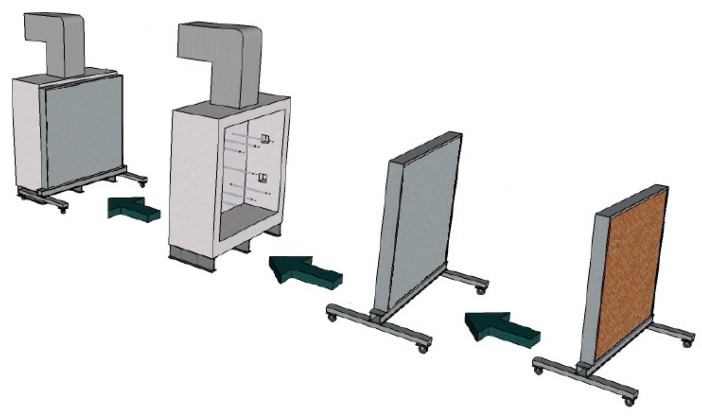
Attaching the samples to the test furnace.

**Figure 5 materials-12-03669-f005:**
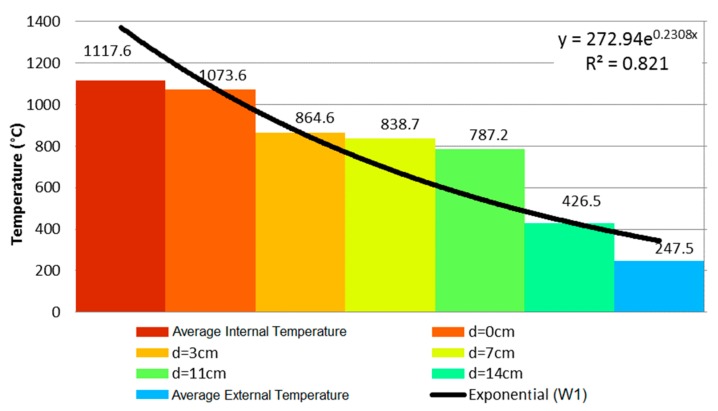
Temperature profile at 240 min—W1.

**Figure 6 materials-12-03669-f006:**
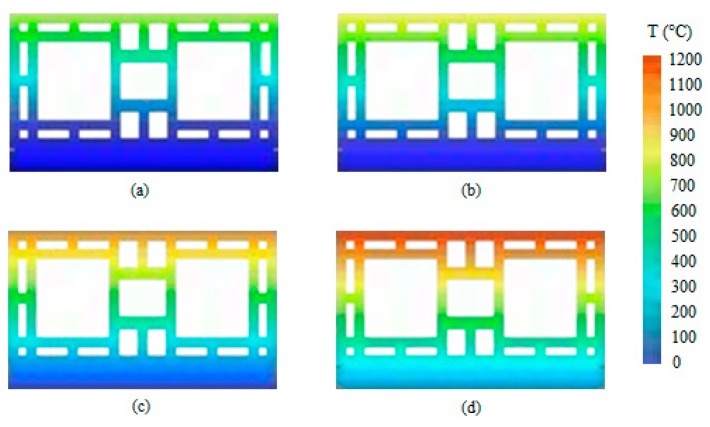
Temperature distribution at (**a**) 30 min, (**b**) 60 min, (**c**) 120 min, and (**d**) 240 min.

**Figure 7 materials-12-03669-f007:**
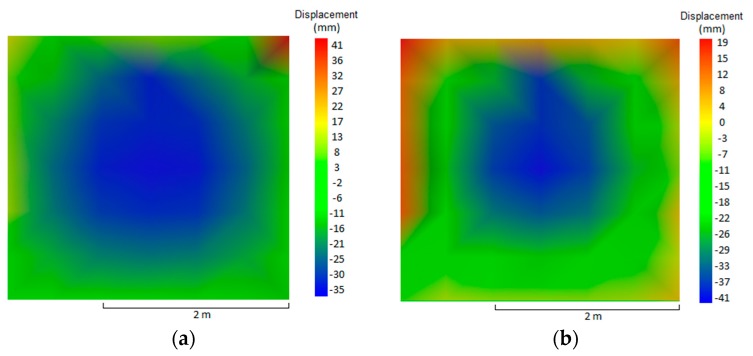
Displacement of the plane at (**a**) 30 min—W1 and (**b**) 150 min—W1.

**Figure 8 materials-12-03669-f008:**
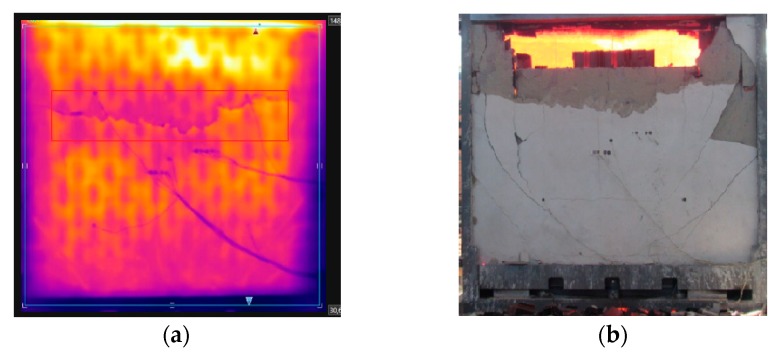
(**a**) Horizontal crack at 52 min and (**b**) loss of structural stability at 102 min.

**Figure 9 materials-12-03669-f009:**
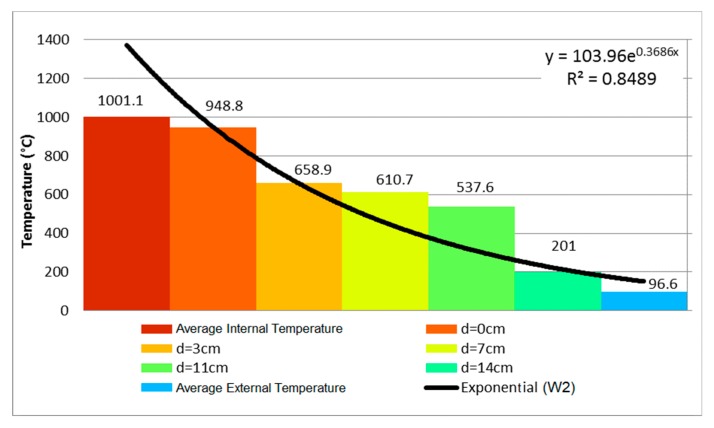
Temperature profile at 102 min—W2.

**Figure 10 materials-12-03669-f010:**
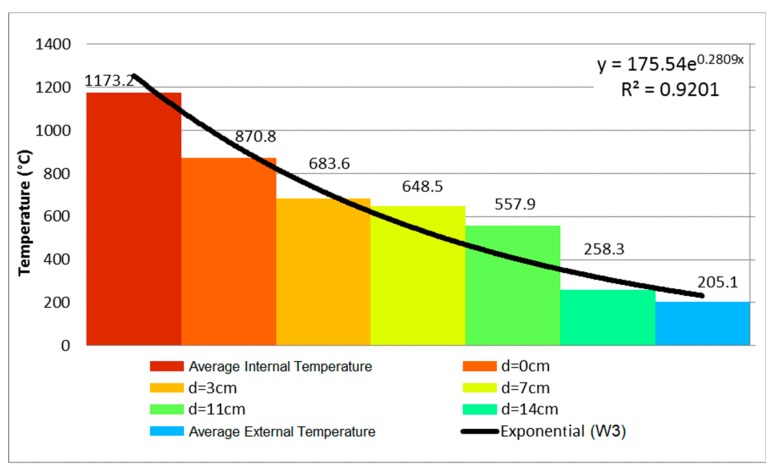
Temperature profile at 240 min—P3.

**Figure 11 materials-12-03669-f011:**
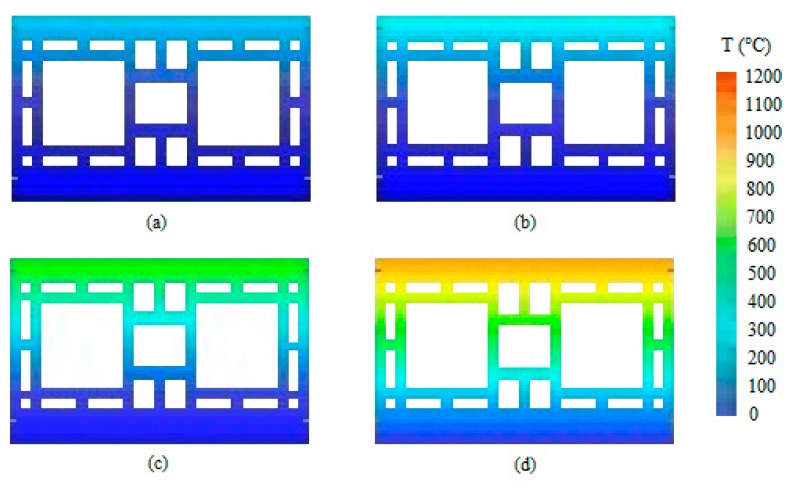
Temperature distribution at (**a**) 30 min, (**b**) 60 min, (**c**) 120 min, and (**d**) 240 min.

**Figure 12 materials-12-03669-f012:**
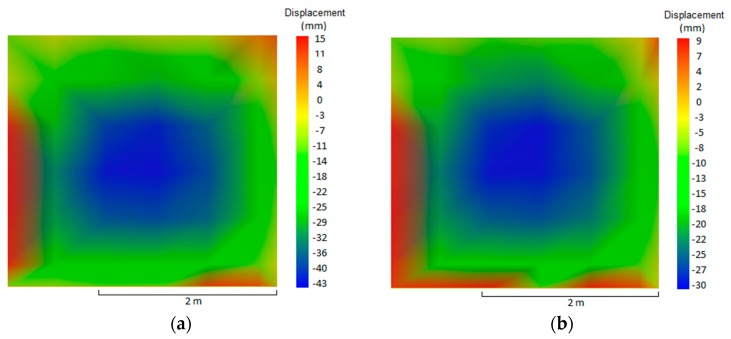
Displacement of the plane at (**a**) 30 min and (**b**) 240 min—W3.

**Figure 13 materials-12-03669-f013:**
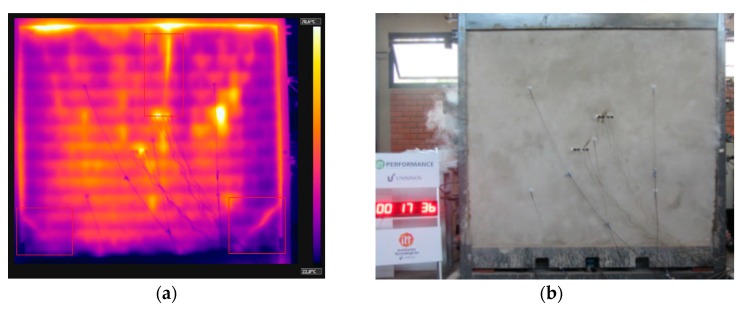
(**a**) Vertical and oblique cracks; (**b**) Smoke coming out of the vertical crack and its surroundings.

**Figure 14 materials-12-03669-f014:**
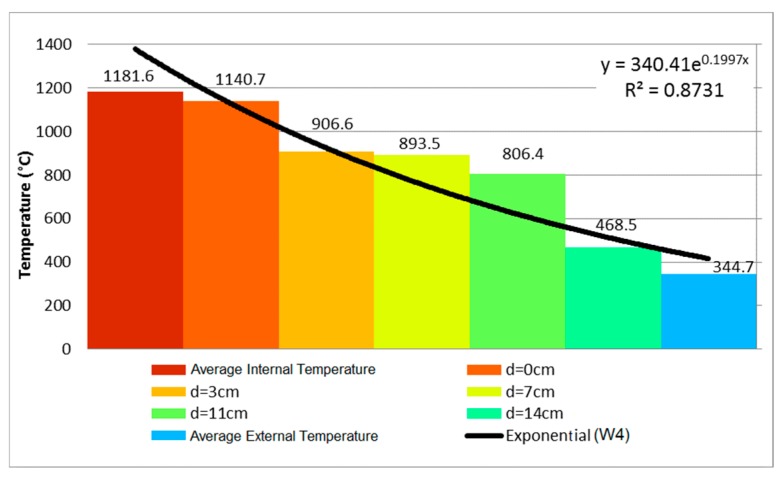
Temperature profile at 240 min—W4.

**Figure 15 materials-12-03669-f015:**
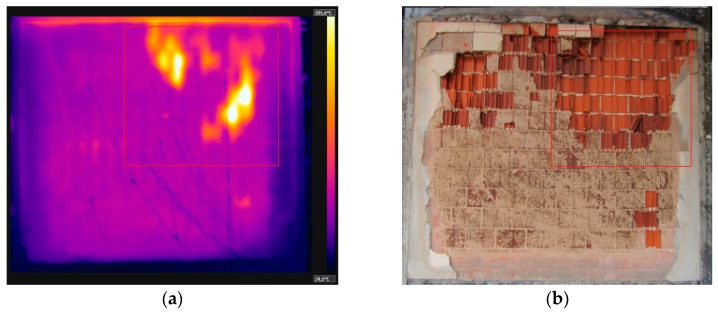
(**a**) Evidence of spalling of the coating on the inner side (**b**) sample after the fire resistance test.

**Figure 16 materials-12-03669-f016:**
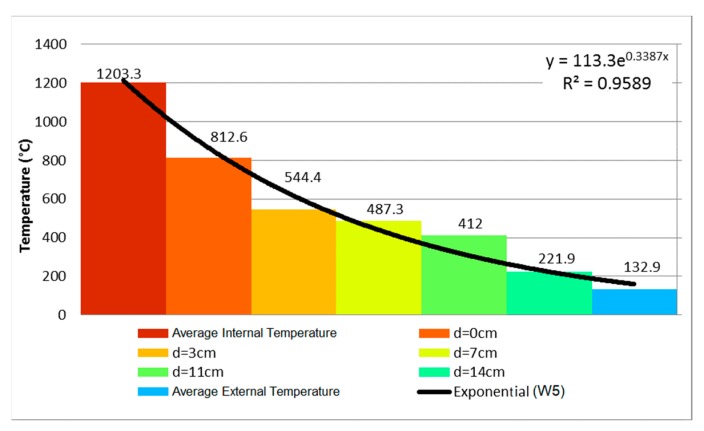
Temperature profile at 240 min—W5.

**Figure 17 materials-12-03669-f017:**
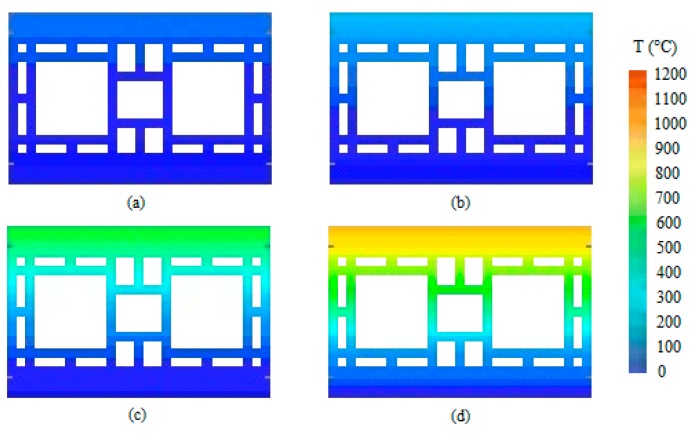
Temperature distribution of the tile at (**a**) 30 min, (**b**) 60 min, (**c**) 120 min and (**d**) 240 min.

**Figure 18 materials-12-03669-f018:**
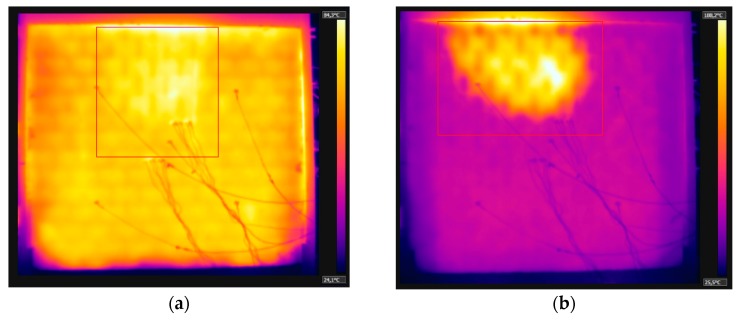
(**a**) Spalling begins—80 min, and (**b**) spalling ends—240 min.

**Figure 19 materials-12-03669-f019:**
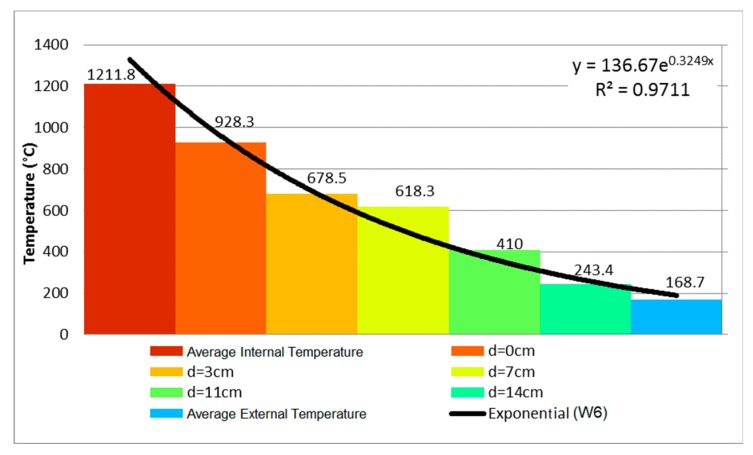
Temperature profile at 240 min—P6.

**Figure 20 materials-12-03669-f020:**
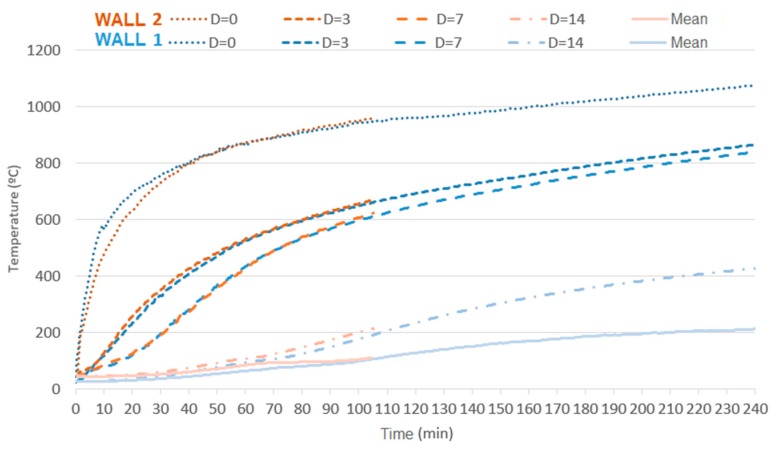
Influence of loading on masonries with no coating on the fire-exposed face.

**Figure 21 materials-12-03669-f021:**
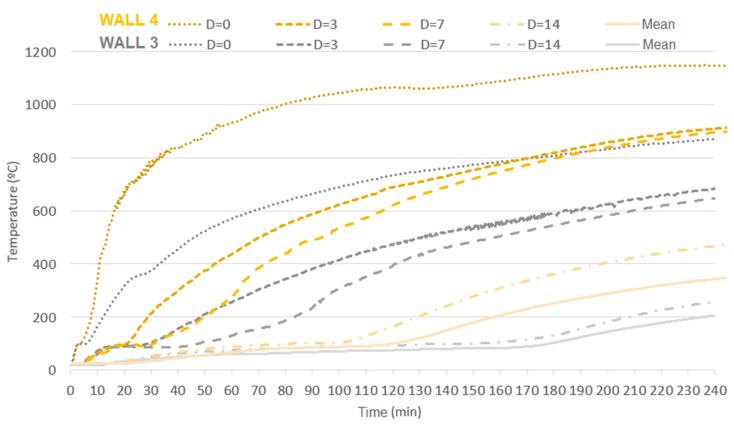
Development of the temperature over time on masonries with 15 mm of coating on the fire-exposed face, without (W3) and with (W4) application of load.

**Figure 22 materials-12-03669-f022:**
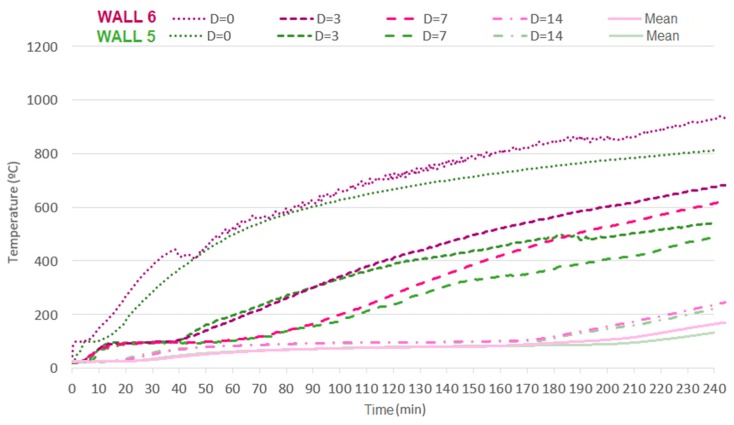
Influence of loading on masonries with 25 mm of coating on the fire-exposed face.

**Figure 23 materials-12-03669-f023:**
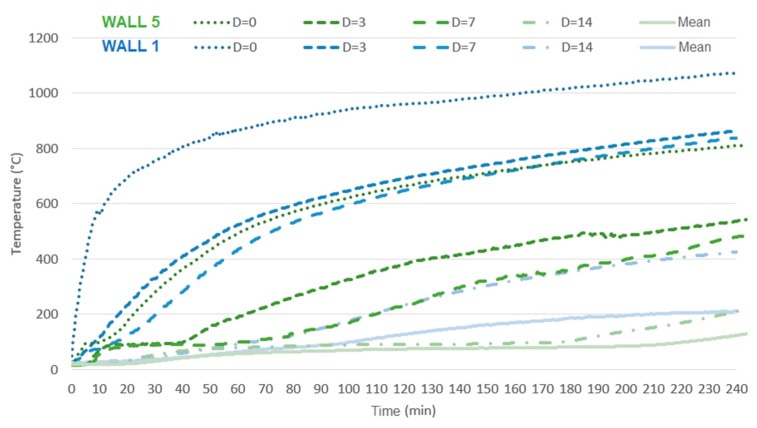
Influence of the addition of 25 mm of coating on the fire-exposed face.

**Table 1 materials-12-03669-t001:** Sample nomenclature.

Samples	Internal Side	External Side	Loading
	Mortar Coating (mm)	Mortar Coating (mm)	(tf/m)
Wall 1 (W1)	None	25	None
Wall 2 (W2)	None	25	10
Wall 3 (W3)	15	25	None
Wall 4 (W4)	15	25	10
Wall 5 (W5)	25	25	None
Wall 6 (W6)	25	25	10

**Table 2 materials-12-03669-t002:** Distribution of fine aggregates—EN 933-1 [31].

Sieve (mm)	Roughcast		Mortar for Laying and Coating	
	Retained (%)	Cumulative (%)	Retained (%)	Cumulative (%)
4.8	4	4	0	0
2.4	6	10	0	0
1.2	11	21	0	0
0.6	13	34	7	7
0.3	36	70	30	37
0.15	28	98	54	91
<0.15	2	100	9	100
Total	100	-	100	-
Maximum size (mm)	4.8		1.2	
Fineness modulus	2.37		1.35	
Specific gravity (g/cm^3^)	2.61		2.63	
Loose bulk density (g/cm^3^)	1.48		1.55	

**Table 3 materials-12-03669-t003:** Characterization of clay tiles according to ASTM C-652 [32].

Tests		Values
**Physical properties**	Length (mm)	291
	Width (mm)	139
	Height (mm)	191
	Warpage of surfaces (mm)	0.5
	Chippage (mm)	0.5
	Total absorption (%)	18
**Mechanic properties**	Compressive strength (MPa)	8.65

**Table 4 materials-12-03669-t004:** Characterization of the mortar for laying and coating.

Characterization Tests		Result
**Fresh state**	Flow level (mm) C-1437 (ASTM, 2015)	265
	Air content (%) C-231 (ASTM, 2017)	3.5
	Density (kg/m^3^) C-185 (ASTM, 2015)	1855.5
**Hardened state**	Compressive strength (MPa) C-109 (ASTM, 2016)	4.7
	Flexural strength (MPa) C-348 (ASTM, 2014)	1.84
	Density (kg/m^3^) C-1754 (ASTM, 2012)	1709
	Capillary water absorption (g/dm^2^·min^½^) C 1403 (ASTM, 2015)	5.8
	Void ratio (%) C-1754 (ASTM, 2012)	28.76

**Table 5 materials-12-03669-t005:** Data compilation.

Sample	Internal Coating (mm)	Loading (tf/m)	Maximum Displacement (mm)	Maximum Average Temperature (°C)	Fire-resistance Rating (FRR)
W1	0	0	41	247.5	131
W2	0	10	32	96.6	81
W3	15	0	43	205.1	208
W4	15	10	21	344.7	142
W5	25	0	39	132.9	240
W6	25	10	24	168.7	221

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
