# Peer review of "Verification of the Influence of Loading and Mortar Coating Thickness on Resistance to High Temperatures Due to Fire on Load-Bearing Masonries with Clay Tiles"

_materials, 2019, doi:10.3390/ma12223669_

Round 1

Reviewer 1 Report

Check English and text editing!

Author Response

Dear Reviewer 1,

The English and text editing were revised.

We appreciate your time and effort in reviewing this manuscript, which will help us improve it to a better scientific level. The manuscript was revised, and changes have taken place according to the valuable suggestions offered by you and by other reviewers.

Therefore, we have sent the revised manuscript, addressed with all issues indicated in the review report, and we believe that the revised version can meet the journal publication requirements.

With respect to the raised concerns, please note the mentioned points that are following discussed.

Reviewer 2 Report

I would suggest the following minor changes:

Fire is more than just temperature. You need to be aware that you are just testing the effect of temperature increase due to a fire, not fire itself. As you speak about masonry I would suggest to include some general references on the study of fire effects on masonry such as:

Gomez-Heras, M., McCabe, S., Smith, B.J., Fort, R. 2009. Impacts of fire on stone-built Heritage: An overview. Journal of Architectural Conservation 15(2), pp. 47-58

Martinho, E., Dionísio, A. 2018. Assessment Techniques for Studying the Effects of Fire on Stone Materials: A Literature Review. International Journal of Architectural Heritage

The authors need to review the formatting of the paper; particularly the reference list.

Author Response

Dear Reviewer 2,

We appreciate your time and effort in reviewing this manuscript, which will help us improve it to a better scientific level. The manuscript was revised, and changes have taken place according to the valuable suggestions offered by you and by other reviewers.

Therefore, we have sent the revised manuscript, addressed with all issues indicated in the review report, and we believe that the revised version can meet the journal publication requirements.

With respect to the raised concerns, please note the mentioned points that are following discussed.

Responses:

Fire is more than just temperature. You need to be aware that you are just testing the effect of temperature increase due to a fire, not fire itself.

We agree with this piece of information, so we have changed the title of the article to clarify the temperature analysis that results from fire actions.

As you speak about masonry I would suggest to include some general references on the study of fire effects on masonry such as:

Gomez-Heras, M., McCabe, S., Smith, B.J., Fort, R. 2009. Impacts of fire on stone-built Heritage: An overview. Journal of Architectural Conservation 15(2), pp. 47-58

Martinho, E., Dionísio, A. 2018. Assessment Techniques for Studying the Effects of Fire on Stone Materials: A Literature Review. International Journal of Architectural Heritage

We believe that these articles are more directly related to buildings made of stone masonry. Therefore, these references have been added as basic bibliography on the subject of masonry in fire conditions. We are thankful for these suggestions.

The authors need to review the formatting of the paper; particularly the reference list.

We have reviewed the formatting and updated the references.

Reviewer 3 Report

Dear Authors!

Thank you for your interesting and niche paper in the field of fire safety of masonry constructions. However, many experiments were done by masonry producers on this topic, almost nothing was published among them. 

The paper is generally well written, results are presented clearly and conclusion is supported by the results and discussion. Here I include some advice and minor text editing changes that I suggest:

Please put space between numbers and units, e.g. instead of "25-mm" or "25mm" write "25 mm". Correct these through the entire manuscript with all numbers and units. You don't need to indicate yourself as the source in the manuscript, especially in tables. Everything is your source if you did not indicate otherwise. Clear "source: made by authors" from the bottom of the tables. Can you write about the conditions during the 56 days long curing process? Temperature, relative humidity conditions? Was the masonry constructions dried out during the curing? Spalling is hugely affected by the moisture content of the construction, therefore, if you may know about the moisture content, please include that also in your manuscript. If you did not measured moisture content, please include this measurement into your experiment design, because it is important information. You can measure relative moisture content as well with non-invasive techniques just to make sure if the construction (especially the applied plaster) dried out evenly.  In Table 2 and 3, you did not include the standards, which you used for the measurements like in Table 4. I find the standards included in the table a piece of useful information, therefore, I advise to include them in all the tables where standardized measurement results are published. In Table 4, please include a horizontal line after the first Density to separate fresh and hard state measurement results. Please include spacing between titles of figures and the text of the manuscript (e.g. spacing between line 189 and 190, etc.) In 2.2.2. section, you write about temperature measurements. The installation of the sensors means the wall are drilled? The drilling holes where the sensors were installed then got some patching? Include these details into the manuscript, too. Where you are writing about the thermal imaging camera, settings and details are missing. You detailed the general capabilities of the thermal imager but did not included the settings. What is the emissivity you used, did you included "atmospheric correction" (air's temperature and relative humidity depending effects) between the surface and the camera and how you included the reflected temperature? Did you measured radiometric video or image sequences? What was the time stepping? In Figure 6 and 11, you show temperature distribution across the horizontal section of the masonry construction. I assume that these are interpolated results based on the measured points or maybe simulated results based on time-dependent heat conduction modeling. Please detail in the manuscript a bit how you created these figures, based on what type of interpolation, etc. If it is created by numerical simulation, then please include the methodology in the manuscript as well. Finally, the tables and figures are varying in their quality of presentation, please unify their look if you can.

Author Response

Dear Reviewer 3,

We appreciate your time and effort in reviewing this manuscript, which will help us improve it to a better scientific level. The manuscript was revised, and changes have taken place according to the valuable suggestions offered by you and by other reviewers.

Therefore, we have sent the revised manuscript, addressed with all issues indicated in the review report, and we believe that the revised version can meet the journal publication requirements.

With respect to the raised concerns, please note the mentioned points that are following discussed.

Responses

Please put space between numbers and units, e.g. instead of "25-mm" or "25mm" write "25 mm". Correct these through the entire manuscript with all numbers and units.

We have corrected the spacing between numbers and units.

You don't need to indicate yourself as the source in the manuscript, especially in tables. Everything is your source if you did not indicate otherwise. Clear "source: made by authors" from the bottom of the tables.

We have removed the sources from the tables.

Can you write about the conditions during the 56 days long curing process? Temperature, relative humidity conditions? Was the masonry constructions dried out during the curing? Spalling is hugely affected by the moisture content of the construction, therefore, if you may know about the moisture content, please include that also in your manuscript. If you did not measured moisture content, please include this measurement into your experiment design, because it is important information. You can measure relative moisture content as well with non-invasive techniques just to make sure if the construction (especially the applied plaster) dried out evenly.

The walls were cured indoors for the period of 56 days, during which theoretical terms suggest the most hydration of mortar and its drying. Since the condition was the same for all samples, it is understandable that an eventful moisture effect would have been noted uniformly by the different walls studied.

In Table 2 and 3, you did not include the standards, which you used for the measurements like in Table 4. I find the standards included in the table a piece of useful information, therefore, I advise to include them in all the tables where standardized measurement results are published.

The standard for Table 3 was already in the text, so it has been added to the title. The standard used to make Table 2 has been added and the numbers of the references have been corrected.

In Table 4, please include a horizontal line after the first Density to separate fresh and hard state measurement results.

Ok, done.

Please include spacing between titles of figures and the text of the manuscript (e.g. spacing between line 189 and 190, etc.)

Ok, done

In 2.2.2. section, you write about temperature measurements. The installation of the sensors means the wall are drilled? The drilling holes where the sensors were installed then got some patching? Include these details into the manuscript, too.

An excerpt has been added to the text to state that the wall was drilled and that there was no need for repairs due to the similarity between the size of the hole and the thermocouple inserted.

Where you are writing about the thermal imaging camera, settings and details are missing. You detailed the general capabilities of the thermal imager but did not included the settings. What is the emissivity you used, did you included "atmospheric correction" (air's temperature and relative humidity depending effects) between the surface and the camera and how you included the reflected temperature? Did you measured radiometric video or image sequences? What was the time stepping?

The thermographic camera used in this study was set for the standard condition. It should be noted that the role of the camera in this study was only to complement the visual analysis, as temperature was measured by the thermocouples.

In Figure 6 and 11, you show temperature distribution across the horizontal section of the masonry construction. I assume that these are interpolated results based on the measured points or maybe simulated results based on time-dependent heat conduction modeling. Please detail in the manuscript a bit how you created these figures, based on what type of interpolation, etc. If it is created by numerical simulation, then please include the methodology in the manuscript as well.

In order to make the gradient presented in the images, the average reading of the thermocouples was performed at every depth as Figure 3 reports. The gradient has 13 bands, varying every interval of 100°C. In the images, the top face of each block points towards the side of the wall that was exposed to high temperatures. The bottom part regards the unexposed face. The images were made in a thermal imaging software.

Finally, the tables and figures are varying in their quality of presentation, please unify their look if you can.

Ok, done.